# A paucigranulocytic asthma host environment promotes the emergence of virulent influenza viral variants

Katina D Hulme[1], Anjana C Karawita[1], Cassandra Pegg[1], Myrna JM Bunte[1], Helle Bielefeldt-Ohmann[1,2,3], Conor J Bloxham[4], Silvie Van den Hoecke[5,6], Yin Xiang Setoh[1,2,7], Bram Vrancken[8], Monique Spronken[9], Lauren E Steele[1], Nathalie AJ Verzele[1], Kyle R Upton[1], Alexander A Khromykh[1,2], Keng Yih Chew[1], Maria Sukkar[10], Simon Phipps[2,11], Kirsty R Short[1,2]*

[1]School of Chemistry and Molecular Biosciences, The University of Queensland, Brisbane, Australia; [2]Australian Infectious Diseases Research Centre, The University of Queensland, Brisbane, Australia; [3]School of Veterinary Science, The University of Queensland, Brisbane, Australia; [4]School of Biomedical Sciences, The University of Queensland, Brisbane, Australia; [5]VIB-UGent Center for Medical Biotechnology, Ghent, Belgium; [6]Department of Biomedical Molecular Biology, Ghent University, Ghent, Belgium; [7]Environmental Health Institute, National Environment Agency, Singapore, Singapore; [8]KU Leuven, Department of Microbiology and Immunology, Rega Institute, Laboratory of Evolutionary and Computational Virology, Leuven, Belgium; [9]Department of Viroscience, Erasmus MC, Rotterdam, Netherlands; [10]Discipline of Pharmacy, Graduate School of Health, University of Technology Sydney, Australia; Woolcock Institute of Medical Research, Sydney Medical School, University of Sydney, NSW, Australia; [11]QIMR Berghofer Medical Research Institute, Brisbane, Australia

*For correspondence:
k.short@uq.edu.au

**Abstract** Influenza virus has a high mutation rate, such that within one host different viral variants can emerge. Evidence suggests that influenza virus variants are more prevalent in pregnant and/or obese individuals due to their impaired interferon response. We have recently shown that the non-allergic, paucigranulocytic subtype of asthma is associated with impaired type I interferon production. Here, we seek to address if this is associated with an increased emergence of influenza virus variants. Compared to controls, mice with paucigranulocytic asthma had increased disease severity and an increased emergence of influenza virus variants. Specifically, PB1 mutations exclusively detected in asthmatic mice were associated with increased polymerase activity. Furthermore, asthmatic host-derived virus led to increased disease severity in wild-type mice. Taken together, these data suggest that at least a subset of patients with asthma may be more susceptible to severe influenza and may be a possible source of new influenza virus variants.

## Introduction

Similar to many other RNA viruses, the influenza virus RNA polymerase has poor proofreading activity, resulting in a high rate of viral mutations (*Van den Hoecke et al., 2015*). Accordingly, even if an influenza virus infection occurs with a defined molecular clone, there will be rapid generation of viral variants that possess one or more mutations relative to the original virus strain. Many viral mutants are of reduced or unaltered virulence compared to the wild-type (WT) parental viral strain (*Van den Hoecke et al., 2015*). However, viral variants that display increased virulence can also emerge.

Modelling studies suggest that there are a higher number of influenza virus variants in individuals that suffer from severe disease (*Reperant et al., 2014*). This may reflect the fact that mutations that are of increased virulence but of reduced fitness are able to replicate in immunocompromised individuals and not in individuals that have a fully competent immune response. Alternatively, in individuals with severe disease there may be additional rounds of virus replication, which represent additional opportunities for viral mutagenesis (*Reperant et al., 2014*). Consistent with these observations, pregnant mice infected with influenza virus have an increased rate of viral mutations compared to their non-pregnant counterparts (*Engels et al., 2017*). Similarly, the impaired interferon response of obese mice is thought to drive the emergence of virulent influenza virus variants (*Honce et al., 2020*). These data raise the question whether asthma, a host co-morbidity frequently associated with an impaired interferon response (*Wark et al., 2005*; *Contoli et al., 2006*), also facilitates the emergence of influenza virus variants.

The impact of pre-existing asthma on the severity of influenza virus infection is controversial. During the 2009 H1N1 pandemic asthma was identified as the single most common underlying medical condition in individuals hospitalised with influenza (*Nguyen-Van-Tam et al., 2010*; *Morris et al., 2012*). Individuals with asthma are known to have a greater risk of (i) admission to the hospital (*Neuzil et al., 2000*; *O'Riordan et al., 2010*; *Van Kerkhove et al., 2011*; *Miller et al., 2008*), (ii) being treated in the intensive care unit (*Bueving et al., 2004*; *Libster et al., 2010*), (iii) requiring a greater number of outpatient visits (*Morris et al., 2012*; *Neuzil et al., 2000*; *Miller et al., 2008*), and (iv) developing pneumonia (*O'Riordan et al., 2010*; *Dawood et al., 2011*) upon influenza virus infection. Whilst some of these observations may be attributable to the fact that influenza virus can exacerbate the symptoms of asthma (*Obuchi et al., 2013*), Dawood and colleagues demonstrated that the majority of patients with asthma were hospitalised for influenza in the absence of an acute asthma attack (*Dawood et al., 2011*). However, a global analysis of pH1N1 cases suggests that once hospitalised, influenza patients with asthma have a greater survival rate than patients with other conditions (*Van Kerkhove et al., 2011*). Consistent with these findings, studies using a mouse model of disease actually suggest that mice with the cardinal features of asthma display *reduced* disease severity following infection with influenza virus (*Furuya et al., 2015*; *Samarasinghe et al., 2014*; *Ishikawa et al., 2012*).

The controversies in the literature regarding the interactions between asthma and influenza appear hard to reconcile. However, it is important to recognise that asthma is a heterogenous disease, consisting of many different phenotypes (*Svenningsen and Nair, 2017*). It is possible that different subtypes of asthma display differing degrees of synergism with influenza virus. For example, the aforementioned mouse studies used either ovalbumin (OVA), fungal allergens, or house dust mite for sensitisation (*Furuya et al., 2015*; *Samarasinghe et al., 2014*; *Ishikawa et al., 2012*), all of which induce an eosinophilic asthma which was characterised by high levels of IgE and eosinophilia. Whilst approximately 40% of adults suffer from this allergic 'eosinophilic asthma', a large percentage of asthma patients suffer from non-allergic asthma (i.e. where asthma is observed in the absence of eosinophilic asthma). One such subtype is paucigranulocytic asthma in which airway remodelling occurs in the absence of eosinophilia or neutrophilia.

Here, we sought to determine the role of paucigranulocytic asthma, using an established experimental mouse model (*Arikkatt et al., 2017*), in the generation of influenza virus variants. We chose to focus on paucigranulocytic asthma for this study as (i) the relationship between this specific phenotype of non-allergic asthma and severe influenza has yet to be studied, (ii) in some populations paucigranulocytic asthma can affect up to 40% of asthmatic patients (*Schleich et al., 2013*), and (iii) paucigranulocytic asthma is associated with an impaired interferon response (*Arikkatt et al., 2017*). We show that mice with a paucigranulocytic asthma phenotype (henceforth referred to as 'asthmatic mice') have increased intra-host influenza viral diversity compared to control mice. Importantly, two of the viral mutations that emerged in asthmatic mice were associated with increased viral replication and polymerase activity. Together, these data provide strong evidence that host–viral interactions are bidirectional and that host co-morbidities can influence the evolution of influenza virus.

## Results

### Pneumonia virus of mice (PVM)-infection of RAGE-deficient mice induces the cardinal features of paucigranulocytic asthma

To investigate the effect of asthma on the severity of influenza we employed a previously established model of paucigranulocytic asthma (*Arikkatt et al., 2017*). In this model, an early-life PVM infection followed by reinfection with PVM later in life in mice deficient in the receptor for advanced glycation endproducts (RAGE), encoded by the *Ager* gene, promotes airway remodelling characteristic of asthma (*Arikkatt et al., 2017*). To confirm this phenotype, the number of mucus-secreting cells and airway smooth muscle mass (ASM) was assessed in PVM-infected RAGE-deficient mice and mock-infected RAGE-deficient mice. Importantly, RAGE-deficient mice that received only an early life PVM infection were not included in these experiments as we have previously shown that an early life PVM infection, followed by a later life infection with influenza A virus (IAV), also results in an asthma-like phenotype (*Arikkatt et al., 2017*). Consistent with our previous findings (*Arikkatt et al., 2017*), an early and later life PVM infection in RAGE-deficient mice resulted in a significant increase in the number of mucus secreting cells and ASM mass (*Figure 1A*). To confirm that the observed structural changes were not associated with a marked pro-inflammatory phenotype or strong interferon response, various cytokines were assessed in the lungs of asthmatic and non-asthmatic mice. Consistent with our previous findings (*Arikkatt et al., 2017*) asthmatic mice had significantly lower pulmonary levels of IFN-γ, CXCL1, TNF-α, CCL2, IL12, IL1b, IL10, and IFN-α compared to non-asthmatic

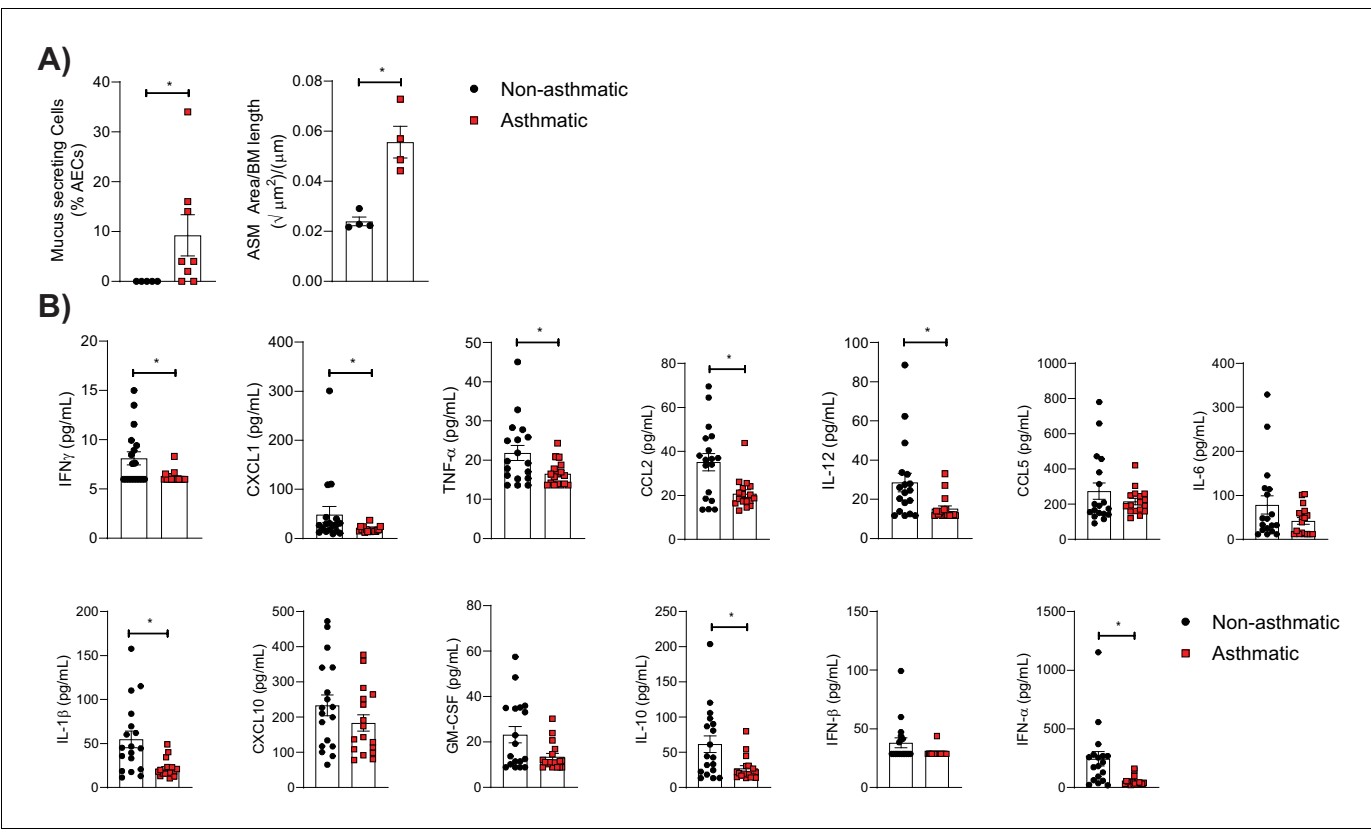

**Figure 1.** Pneumonia virus of mice (PVM)-infection (at 7 days of age and reinfection at 35 days of age) of RAGE-deficient mice induces the cardinal features of asthma. (**A**) (Left) Percentage of airway epithelial cells (AECs) producing mucus was quantified by immunohistochemistry at 7 days post-reinfection (i.e. 42 days of age) from uninfected RAGE-deficient mice (non-asthmatic) and infected RAGE-deficient mice (asthmatic). (Right) Lung sections from non-asthmatic and asthmatic mice at 42 days of age were stained for smooth muscle actin by immunohistochemistry and the airway smooth muscle (ASM) area was calculated relative to the basement membrane (BM) length of small airways. (**B**) Cytokine levels in the lung of RAGE-deficient mice 7 days post-PVM or mock infection (i.e. 42 days of age). Statistical analysis was performed as described in Materials and methods with *p≤0.05. Data are pooled from at least two independent experiments and shown as mean ± SEM.

control mice (*Figure 1B*). These data indicate that this is an appropriate mouse model to study the effects of asthma, and an impaired interferon response, on the emergence of influenza virus variants.

## Mice with the features of paucigranulocytic asthma had increased IAV severity

To the best of our knowledge the association between non-allergic asthma and influenza has not been investigated. Accordingly, we sought to investigate the severity of IAV infection in the non-allergic model of asthma described above. Asthmatic mice infected with IAV lost significantly more body weight over time compared to non-asthmatic control mice (*Figure 2A*). This was associated with increased IAV replication in the lungs of asthmatic mice at 6 days post-infection (d.p.i) (*Figure 2B*). While there was an increase in the histopathology score of mock infected asthmatic mice compared to non-asthmatic mice, there was no observable difference following IAV infection (*Figure 2C*). Furthermore, asthmatic mice showed a significantly impaired type I interferon response, with a downregulation of both IFN-α and interferon inducible MX1 following IAV infection (*Figure 2D*).

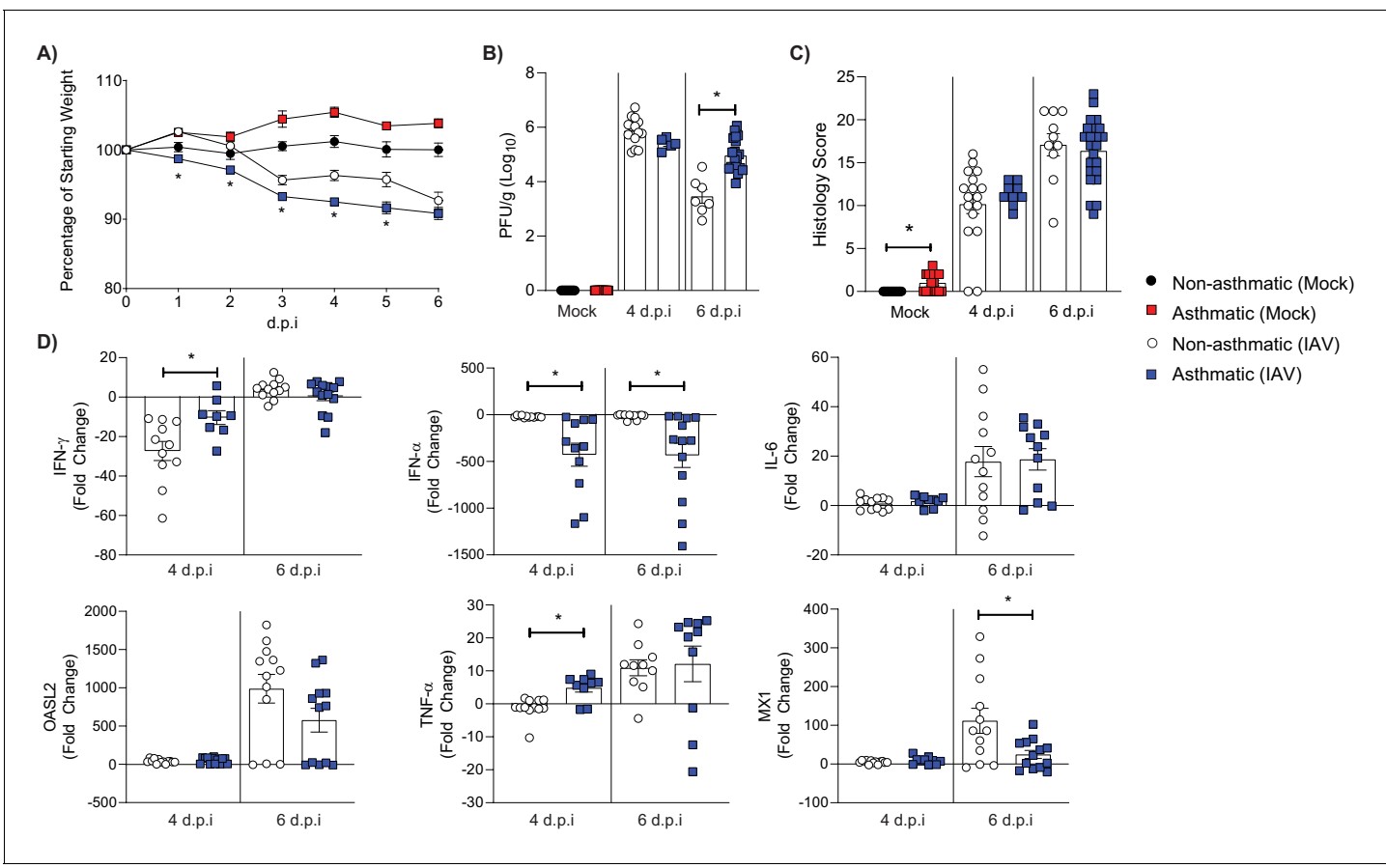

**Figure 2.** RAGE-deficient asthmatic mice experience more severe influenza than non-asthmatic mice. RAGE-deficient non-asthmatic and asthmatic mice were infected with 100 (PFU) of H1N1/Auck/09. Mock control groups received PBS only. (**A**) Percentage weight loss of influenza A virus (IAV) or mock infected mice. Weights are displayed as percentage of a mouse's weight at the time of infection. Each data point represents mean ± SEM, with at least 17 mice per group. (**B**) Viral titres present in lung homogenate at 4 and 6 days post-infection (d.p.i.) in IAV and mock-infected mice. (**C**) Histopathology score of lung sections for vascular changes, bronchitis, interstitial inflammation, alveolar inflammation, pneumocyte hypertrophy, and pleuritis combined. (**D**) Cytokines in lung homogenate at 4 and 6 d.p.i. Data are normalised to *Gapdh* expression and fold change was calculated using the ΔΔCt method, expressed relative to mock infected mice. Statistical analysis was performed as described in Materials and methods with *p≤0.05, comparing non-asthmatic IAV to asthmatic IAV. Data are pooled from at least two independent experiments and shown as mean ± SEM (**A–D**), with a data point representing one mouse (**B–D**).

## An asthmatic host environment led to increased non-synonymous IAV mutations

Having established that the asthma phenotype employed herein was associated with increased IAV severity we next sought to determine if asthmatic mice had an increased number of IAV variants. To do so, we used deep sequencing to analyse viral sequences isolated from the lungs of infected mice at 4 and 6 d.p.i. Mutations were only included in the analysis if they constituted >1% of the total viral population and were detected in a minimum of 1000 reads following quality control. Shannon's entropy was determined for each of the eight influenza genome segments in both 4 and 6 d.p.i. samples (*Figure 3*). To account for any bias from difference in sequencing coverage, the read count for each segment was analysed. No significant difference was observed in sequence coverage for any of the influenza virus genes between the two treatment groups (*Figure 3—figure supplement 1*). Asthmatic host-derived samples showed significantly more within-host viral diversity of the PB1 segment at 6 d.p.i. (*Figure 3B*). To determine if the mutations were associated with a specific section of the PB1 gene, their frequency was visualised against the nucleotide position, where no discernible trend was observed (*Figure 4*).

To determine if any of the mutations detected in the asthmatic mice could have contributed to the increased IAV severity observed, we focused on unique (occurring specific to either non-asthmatic or asthmatic hosts) non-synonymous mutations within the PB1 segment, as these were the most likely to be of functional significance. This produced a list of 404 unique mutations across all groups (*Supplementary file 2*). We then refined our analysis to only include non-synonymous mutations that were detected in two or more mice. Overall, there was an increase in unique mutations

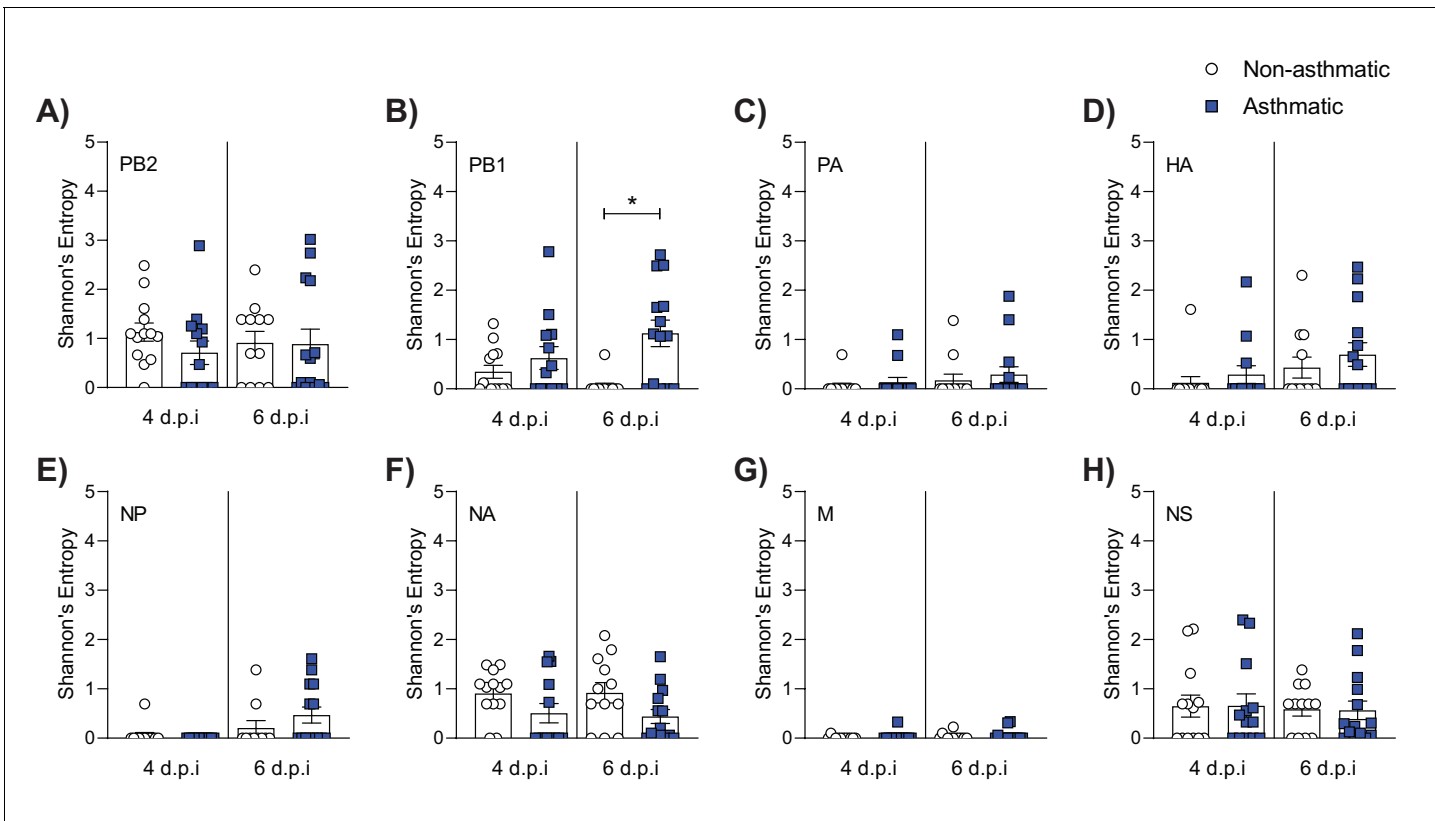

**Figure 3.** Asthmatic hosts produced more diverse viral variants in PB1. Influenza viral RNA isolated from the lung tissue of asthmatic (n = 27) and non-asthmatic (n = 25) mice was analysed for viral variants at 4 and 6 days post-infection (d.p.i.). (A–H) Measurement of alpha diversity presented as Shannon entropy of viral mutations at a frequency above 1% by genome segment at the nucleotide level. Statistical analysis was performed as described in Materials and methods with *p≤0.05. Data are shown as mean ± SEM.

The online version of this article includes the following figure supplement(s) for figure 3:

**Figure supplement 1.** Read depth of the eight Influenza genome segments from *Ager*-/- mice.

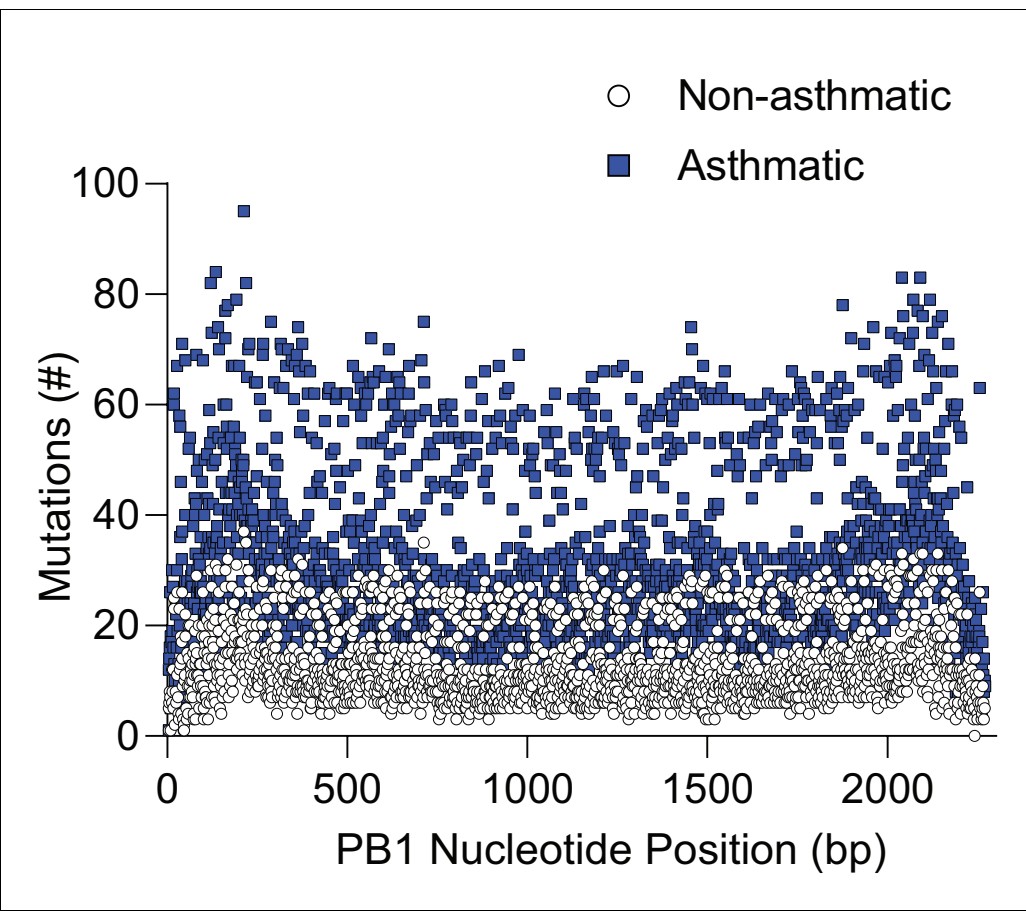

**Figure 4.** Asthmatic hosts produced more single nucleotide variants (SNVs) in the PB1 genome segment. Influenza viral RNA isolated from the lung tissue of asthmatic (n = 27) and non-asthmatic (n = 25) mice was analysed for viral variants. Each data point represents the number of viral variants detected at that nucleotide position within the viral genome PB1 segment. bp: base pair.

detected in asthmatic compared to non-asthmatic host samples, with 32 and 5, respectively (*Table 1*). Interestingly, the bulk of the mutations were focused around the priming loop, thumb, and C-terminal extension of PB1.

We next sought to determine if any of these IAV mutations detected in mouse samples had also been reported in pH1N1 IAV patients with asthma. To do so, post-2009 H1N1 viral sequences from patients that listed 'asthma' as a medical co-morbidity (n = 51) were obtained from the Influenza Research Database (IRD). This was compared to viral sequences obtained from 51 age and sex-matched pH1N1 patients on the IRD with no medical co-morbidities. Strikingly, of the unique mouse single nucleotide variants, two that were unique to asthmatic mouse samples were also detected in clinical samples at a higher frequency in asthmatic patient samples (*Table 1*). Interestingly, of the mutations identified in the clinical consensus sequence samples, the K691R mutation reached a high enough frequency in two asthmatic mouse samples to alter the consensus sequence.

## Asthmatic host-derived samples showed increased replication kinetics *in vitro*

Having established that an asthmatic host generated more viral variants, specifically within PB1, we next sought to assess if these mutations influenced the replicative ability of asthmatic and non-asthmatic mouse-derived IAV samples *in vitro*. To do so, we infected Madin–Darby canine kidney (MDCK) cells, human alveolar basal epithelial cells (A549 cells), and Immortalized Mouse Mammary Epithelial Cells (iMMEC) at a multiplicity of infection (MOI) of 0.01 with either lung homogenates derived from three

**Table 1.** Frequency of exclusive non-synonymous viral mutations detected in PB1 at 4 and 6 days post-influenza A virus (IAV) inoculation and their frequency in clinical samples.

| Non-asthmatic | | | | Asthmatic | | | | Clinical samples (n = 51 per group) | | |
|---|---|---|---|---|---|---|---|---|---|---|
| Day 4 (n* = 13) | | Day 6 (n = 12) | | Day 4 (n = 13) | | Day 6 (n = 14) | | Non-asthmatic | Asthmatic | p-Value[‡] |
| | | M1I | 50%[†] | | | | | 0% | 0% | |
| | | M1L | 17% | | | | | 0% | 0% | |
| | | | | | | D2H | 14% | 0% | 0% | |
| | | | | | | N77I | 14% | 0% | 0% | |
| | | | | | | P74A | 14% | 0% | 0% | |
| | | | | | | E75D | 29% | 0% | 0% | |
| | | | | | | E75V | 14% | 0% | 0% | |
| | | | | | | D76A | 14% | 0% | 0% | |
| | | | | | | D76H | 14% | 0% | 0% | |
| | | | | | | N77V | 14% | 0% | 0% | |
| | | | | | | E78G | 14% | 0% | 0% | |
| | | | | | | S80H | 14% | 0% | 0% | |
| | | | | | | I205M | 14% | 0% | 0% | |
| | | | | | | I248V | 14% | 0% | 0% | |
| | | | | | | M616I | 14% | 0% | 0% | |
| | | | | | | R621P | 14% | 0% | 0% | |
| | | | | | | C625S | 14% | 0% | 0% | |
| | | | | | | A652P | 14% | 0% | 0% | |
| | | | | | | S654N | 14% | 4% | 18% | 0.0514 |
| | | | | | | Y657H | 14% | 0% | 0% | |
| | | N671H | 17% | | | R670S | 14% | 0% | 0% | |
| R672P | 15% | | | | | | | 0% | 0% | |
| | | S673A | 17% | | | | | 0% | 0% | |
| | | | | I682F | 15% | | | 0% | 0% | |
| | | | | D685E | 15% | | | 0% | 0% | |
| | | | | | | D685H | 14% | 0% | 0% | |
| | | | | | | M688I | 14% | 0% | 0% | |
| | | | | | | Y689D | 21% | 0% | 0% | |
| | | | | | | Y689R | 14% | 0% | 0% | |
| | | | | Q690S | 15% | | | 0% | 0% | |
| | | | | K691R | 15% | | | 4% | 16% | 0.092 |
| | | | | | | C692S | 14% | 0% | 0% | |
| | | | | | | C693N | 14% | 0% | 0% | |
| | | | | | | N694R | 14% | 0% | 0% | |
| | | | | L695I | 15% | | | 0% | 0% | |
| | | | | | | R727T | 14% | 0% | 0% | |

*n = Total number of mouse samples.

[†]Percentage of mice with identified mutation.

[‡]Statistical comparisons were made using a two-sided fisher's exact test.

asthmatic mice or three non-asthmatic mice. The viruses in the asthmatic lung homogenates replicated to significantly higher titres than those from non-asthmatic lung homogenates at 24 hr post-infection in MDCK cells (*Figure 5A*). A similar trend was observed in mouse and human epithelial cells (*Figure 5B and C*) but with a delayed rate of replication.

Interestingly, comparing two of the viral samples pre and post *in vitro* replication in MDCKs we did not detect a change in the consensus sequence of the daughter population after *in vitro* replication compared to the consensus sequence of the parent. In terms of minority viral populations, in one daughter population no viral variants (with non-synonymous mutations) were detected that

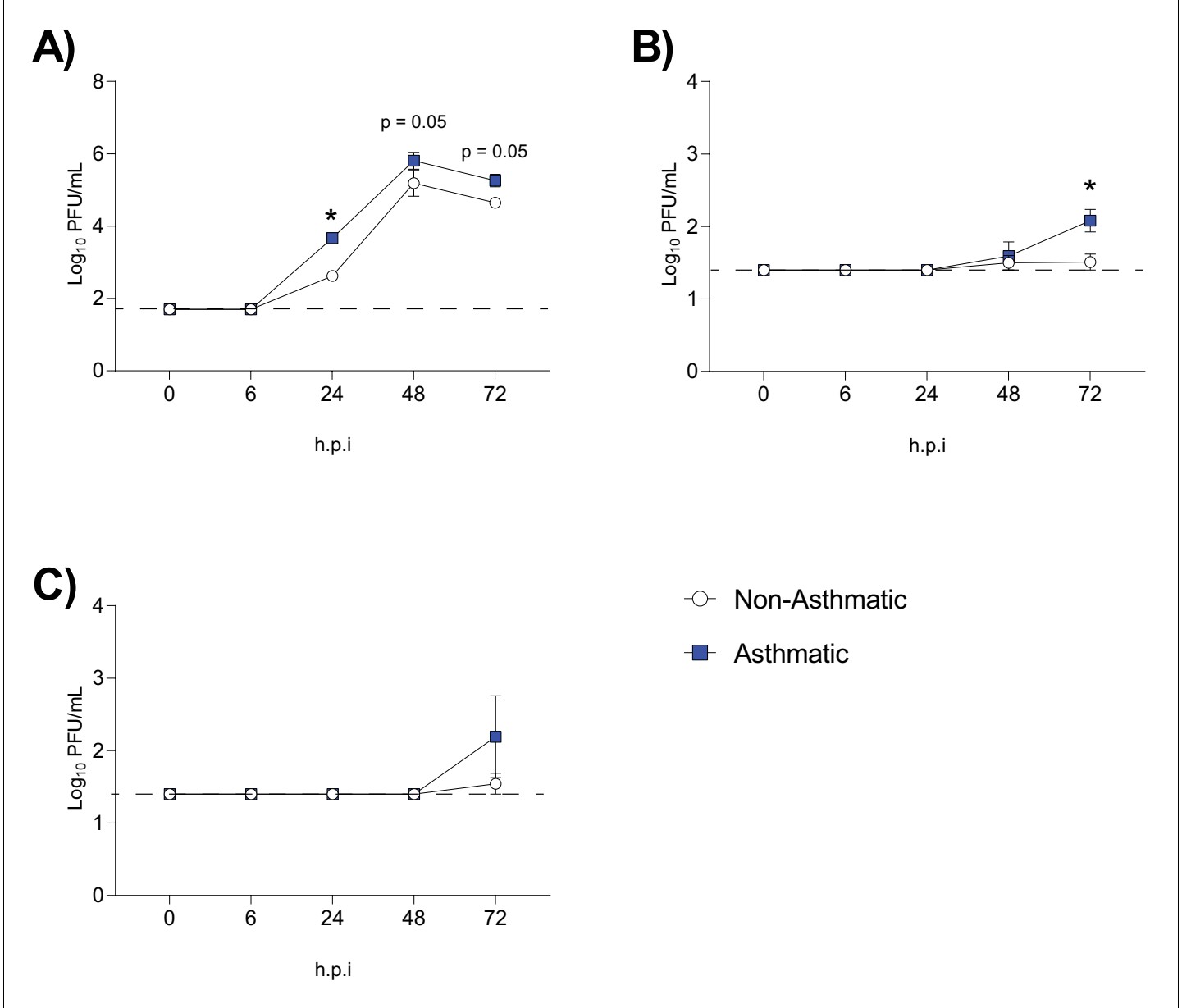

**Figure 5.** Increased replication of influenza A virus (IAV) was observed in samples derived from asthmatic hosts *in vitro*. IAV isolated from asthmatic hosts replicates more quickly than non-asthmatic hosts in (A) Madin–Darby canine kidney (MDCK) cells, (B) Immortalized Mouse Mammary Epithelial Cells (iMMEC), and (C) adenocarcinomic human alveolar basal epithelial cells (A549 cells). Each cell type was infected with the indicated viruses at a multiplicity of infection (MOI) of 0.01 (n = 3 wells/virus/time point). Mock infected data not shown. Data were analysed using a one-way ANOVA with Holm–Sidak's multiple comparisons test and are represented as mean ± SEM. *p<0.05. Dashed line indicates the detection limit of the assay.

constituted between 5 and 49% of the viral population. In the second daughter population, one non-synonymous mutation (PB1 I205M) was detected that constituted 28% of the viral population. However, this mutation was present in the consensus sequence of the parent isolate, albeit at a higher percentage (95%). Together, these data suggest that the increased replication of viral isolates from asthmatic mice was not associated with a large number of additional *in vitro* adaptations.

## PB1 C693Y and P651Q may increase the stability of the influenza RNA polymerase protein dimer

We next sought to elucidate if any of the observed mutations identified in the asthmatic group could be responsible for increased viral replication. To do so, four mutations were selected for structural modelling; two that appeared in multiple mouse samples as well as clinical samples (S694N and K691R), and two that were found in areas likely to affect function (C693Y and P651Q) (*Supplementary file 2*).

None of the mutations were predicted to have a major destabilising effect and all mutations maintained the secondary structures within PB1. The Foldx results predicted S654N (ΔΔG −0.07 kcal/mol) and K691R (ΔΔG 0.43 kcal/mol) to be neutral and C693Y (ΔΔG −1.86 kcal/mol) and P651Q (ΔΔG −1.78 kcal/mol) to be stabilising as defined by *Studer et al., 2014*. As the C-terminus extension interacts with PB2, Foldx was used to highlight potential interactions between the C-terminal mutated residues and residues of PB2. The structural modelling of C693Y identified a side chain–side chain interaction in the mutant between PB1 C693Y and PB2 K32 in the third α-helix (*Figure 6A*). In contrast, PB1 WT C693 was only predicted to interact with resides within the first PB1 α-helix. Together, these data suggest that the C693Y mutation identified exclusively in asthmatic hosts could alter the interaction between PB1 and PB2 within the polymerase heterotrimer. No close contacts were identified between K691R and residues of PB2 (*Figure 6A*). S654N and P651Q reside within the conserved anti-parallel β-hairpin with P651Q sitting in the tip of the β-hairpin (*Figure 6B*). Both mutations within the β-hairpin changed the orientation of the residue, which could suggest a functional change.

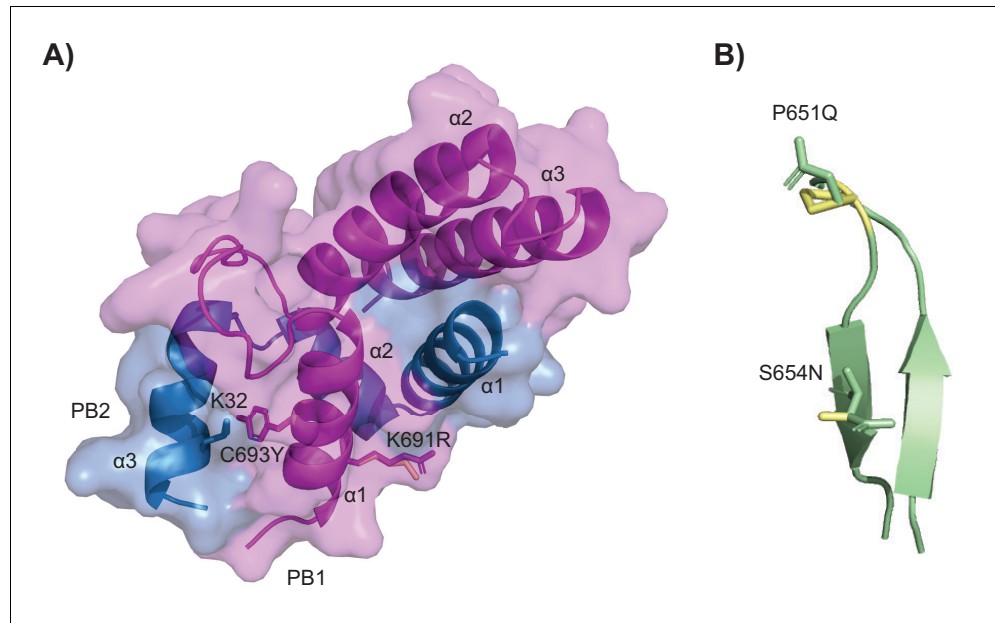

**Figure 6.** Crystal structures of the mutations within PB1. (**A**) Mutations C693Y and K691R. The molecule is displayed in surface view with cartoon representation of the protein backbone (PB1 magenta and PB2 sky blue). The α-helices are labelled and PB1 C693Y and K691R and PB2 K32 are shown as sticks in the respective colours of the subunits with the wild-type (WT) residues in pale yellow. (**B**) Mutations P651Q and S654N within the conserved β-hairpin (cartoon representation in pale green). The sites of mutations are shown as sticks with the WT residues in pale yellow.

## PB1 C693Y and P651Q increase viral polymerase activity

We next sought to investigate the individual effect of the PB1 C693Y and P651Q mutations on viral polymerase activity using a minigenome assay, given their predicted increase in stability. Both the PB1 C693Y and P651Q mutations within the PB1 gene increased polymerase activity when compared to the WT virus polymerase (*Figure 7*). Together these data suggest that the C693Y and P651Q mutations identified exclusively in asthmatic hosts may alter the interaction between PB1 and PB2 in a stabilising manner, resulting in increased polymerase activity.

## Asthmatic host-derived IAV exhibits greater disease severity and viral diversity upon infection of a WT host

Vulnerable patient groups may be a source for the emergence of new viral strains (*Honce et al., 2020*). As such, we next sought to determine if asthmatic host-derived IAV would induce greater disease severity in WT mice. We inoculated WT mice with either asthmatic or non-asthmatic host-derived IAV and monitored them for 4 days, at which point they were euthanised. Mice infected with asthmatic host-derived IAV had significantly more weight loss (*Figure 8A*), lower blood oxygen saturation (*Figure 8B*), and higher viral titres (*Figure 8C*) compared to mice infected with non-asthmatic host-derived IAV. There was also a non-significant trend towards increased viral mRNA present (*Figure 8D*). Mice infected with asthmatic host-derived IAV had a significant upregulation in various anti-viral and pro-inflammatory cytokines (*Figure 8E*).

Furthermore, upon infection of a non-asthmatic host with non-asthmatic-derived IAV, there is an observable divergence in the PB1 haplotype differences as the virus continues to adapt to its host (*Figure 9A*). By contrast, upon infection of a non-asthmatic host with asthmatic-derived IAV, there is a consolidation of the PB1 mutations that arose during the initial infection (*Figure 9B*). Together these data suggests that asthmatic hosts could facilitate the emergence of new strains that have increased pathogenicity.

## Discussion

Asthma was identified as the most prevalent underlying host condition associated with hospitalisation for pH1N1 in 2009 (*Nguyen-Van-Tam et al., 2010*; *Morris et al., 2012*). However, the impact of pre-existing asthma on the severity of influenza remains unclear (*O'Riordan et al., 2010*; *Dawood et al., 2011*; *Furuya et al., 2015*; *Samarasinghe et al., 2014*; *Ishikawa et al., 2012*). Mouse studies modelling allergic asthma have suggested that asthma may mediate partial protection from severe influenza (*Furuya et al., 2015*; *Samarasinghe et al., 2014*; *Ishikawa et al., 2012*; *Doorley et al., 2017*). Here, using our previously established mouse model of non-allergic paucigranulocytic asthma (*Arikkatt et al., 2017*), we show that the impaired interferon response (characteristic of this subtype) is associated with increased influenza severity. These data suggest that further research, both clinical and pre-clinical, into the relationship between non-allergic asthma and influenza virus severity is warranted.

Recent data suggest that an impaired interferon response correlates with an increase in the emergence of influenza viral variants (*Engels et al., 2017*; *Honce et al., 2020*). Engels and colleagues demonstrated that an allogenic

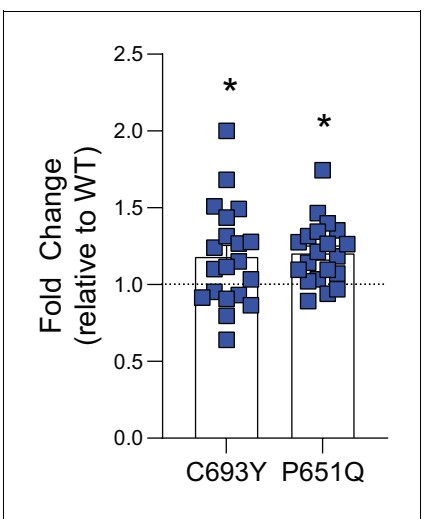

**Figure 7.** The PB1 C694Y and P651Q mutations showed increased polymerase activity compared to the wild type (WT). Influenza RNA polymerase-driven activity is increased compared to WT in HEK 293 T cells transfected with either the C693Y or P651Q mutation in PB1. Data are pooled from four independent experiments (with five biological replicates per group, per experiment). Results are presented as a fold change relative to WT. Statistical significance was determined as described in the Materials and methods. Data shown as mean ± SEM. *p<0.05.

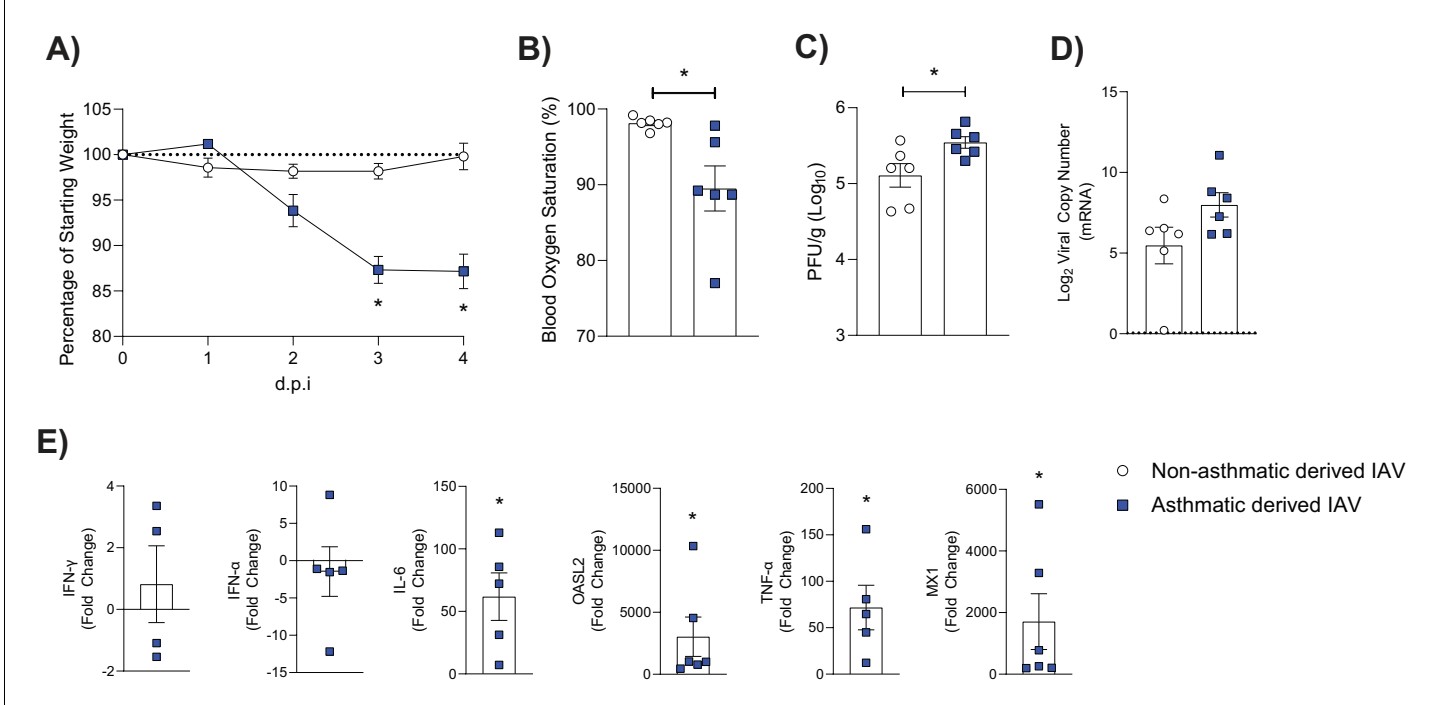

**Figure 8.** Mice infected with asthmatic host-derived virus experience more severe influenza than mice infected with non-asthmatic host-derived virus.

C57BL/6 mice were infected with 2000 PFU of either asthmatic or non-asthmatic host-derived influenza A virus (IAV). (A) Percentage weight loss of mice following IAV infection. Weights are displayed as percentage of a mouse's weight at the time of infection. Each data point represents mean ± SEM (n = 6 per group). (B) Percentage blood oxygen saturation of mice at 4 days post-infection (d.p.i.). (C) Viral titres present in lung homogenate at 4 d.p.i. in IAV-infected mice. (D) Viral mRNA detected by qPCR from murine lung homogenate at 4 d.p.i. Dashed line indicates the detection limit. (E) Cytokines in lung homogenate at 4 d.p.i. Data are normalised to *Gapdh* expression and fold change was calculated using the ΔΔCt method, expressed relative to samples infected with non-asthmatic-derived virus. Statistical analysis was performed as described in Materials and methods with *p≤0.05. Data are shown as mean ± SEM.

pregnant host environment fails to activate the innate immune response, including that of type I interferons during IAV infection (*Engels et al., 2017*). The failure to mount a robust immune response was associated with an increased viral load and a higher emergence of viral variants (*Engels et al., 2017*). More recently, Honce and colleagues demonstrated that the decreased type I interferon response within an obese micro-environment corresponded to increased viral replication and enhanced virulence compared to WT mice (*Honce et al., 2020*). Consistent with these findings, we observed increased viral variants in mice with the cardinal features of paucigranulocytic asthma, particularly within PB1. The absence of an early immune response in the asthmatic mice, that is, lack of type I interferon expression, may allow for the emergence of viral variants that would otherwise be suppressed in a healthy host. Further to this, non-asthmatic host-derived viruses showed host adaption and little sign of morbidity in WT mice, while asthmatic host-derived virus resulted in increased morbidity. This suggests that viral variants born from an asthmatic host could (transmission permitting) lead to the emergence of more pathogenic viral strains.

The influenza A viral RNA-dependent RNA polymerase (RdRp) forms dimers of heterotrimers consisting of PA, PB1, and PB2 (*Fan et al., 2019*), with areas of inter-subunit contact stabilising the trimers (*Pflug et al., 2014*; *Sugiyama et al., 2009*). Intra-host viral mutations in this complex may be deleterious, neutral, or advantageous in terms of viral pathogenesis. The β-hairpin within the PB1 thumb domain is responsible for viral replication initiation, and deletions in this region impair viral replication and growth (*Oymans and te Velthuis, 2018*).

PB1 S654N (a mutation identified in asthmatic clinical samples and those from asthmatic mice) and PB1 P651Q (a mutation identified in asthmatic mice) reside within this priming loop. Interestingly, the P651 residue has previously been shown to be the primary amino acid involved in coordinating terminal initiation during replication (*Te Velthuis et al., 2016*). Glutamine residues have been documented to interact with nucleotide bases (*Luscombe et al., 2001*), and given the position of PB1 P651Q, it is possible that it alters the interaction with the 3′ sugar-base of the vRNA or is

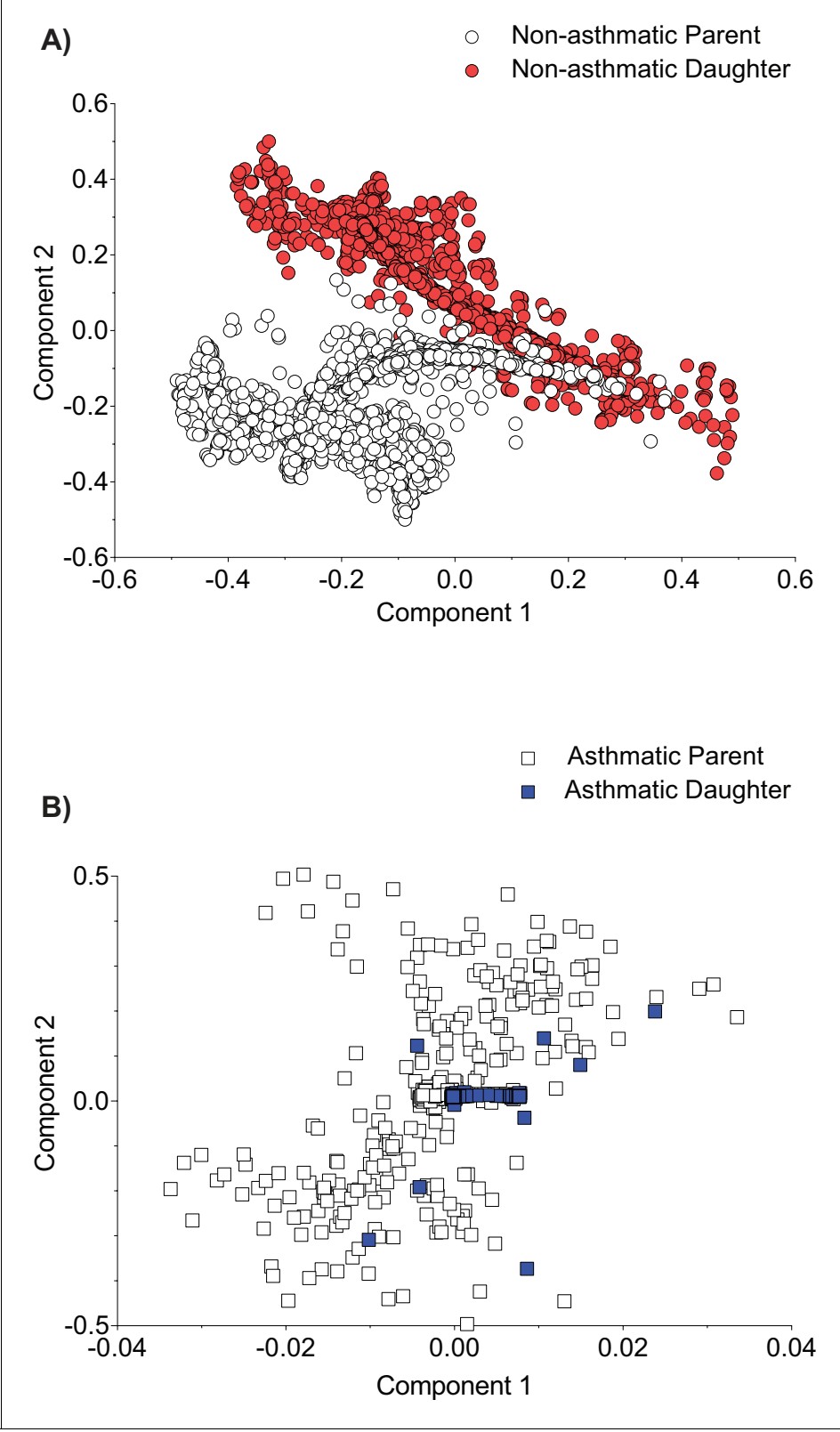

**Figure 9.** Asthmatic host-derived virus shows consolidation of PB1 viral variants population across parent and daughter. C57BL/6 mice were infected with 6 days post-infection (d.p.i.) murine lung homogenate from either non-asthmatic or asthmatic hosts. Multiple dimension scaling (MDS) analysis of the PB1 gene in (**A**) non-asthmatic-derived virus and (**B**) asthmatic-derived virus (each data point represents an individual haplotype). Data is

*Figure 9 continued on next page*

*Figure 9 continued*

representative of six samples per group. Parent: virus derived from either asthmatic or non-asthmatic *Ager*[-/-] host. Daughter: virus isolated from C57BL/6 mice lung following infection with host-derived virus.

beneficial to structural conformation during the initiation process. Our structural analysis of PB1 P651Q predicted a stabilising effect, which was consistent with an increase in polymerase activity relative to the WT. C693Y and K691R (both of which were identified in asthmatic mice) reside in the first α-helix of the C-terminus, which constitutes the major binding region with the N-terminus of PB2 and has an overall major stabilising effect on the heterotrimer (*Pflug et al., 2014*; *Sugiyama et al., 2009*; *Te Velthuis and Fodor, 2016*; *Sugiyama et al., 2009*; *Poole et al., 2007*). Deletion of any of the three α-helices within the C-terminus abolishes viral gene expression (*Poole et al., 2007*). Double and single mutations in the C-terminus of PB1 alter viral RNA yield (*Sugiyama et al., 2009*). Our structural analysis of PB1 C693Y predicted side chain interactions between the mutated residue and PB2 K32 which may increase the stability of the inter-subunits and, in turn, affect polymerase activity. Thus cumulatively, the data presented here suggest that the PB1 C693Y and P651Q mutations detected in asthmatic mice resulted in increased viral replication in various epithelial cell lines, and increased polymerase activity. It is well established that single nucleotide variants (SNVs) can drastically change the replication fidelity and pathogenicity of a virus (*Lin et al., 2019*; *Pauly et al., 2017*). As such, future research would benefit from more in-depth analysis of the effect of C693Y or P651Q within PB1 on the fidelity of the RdRp. Nevertheless, our results should be taken as proof of principal that an asthmatic host may be more likely to produce more virulent viral variants – a finding that holds wide consequences for the emergence of new viral strains.

It is important to recognise that there are several limitations associated with this study. Firstly, paucigranulocytic asthma remains a poorly understood and poorly characterised phenotype of asthma (*Tliba and Panettieri, 2019*). Accordingly, we are not aware of any other mouse model of paucigranulocytic asthma that could be used to validate these results. Similarly, there was insufficient information associated with the clinical sequences used herein to determine which asthma phenotypes each patient had. Indeed, we cannot eliminate the possibility that all patients suffered from allergic, rather than non-allergic, asthma. The clinical sequences analysed in the present study were also obtained as consensus sequences, and do not account for minority viral variant populations present in any individual. Additionally, only a single measure of genetic diversity was used (Shannon entropy) (*Honce et al., 2020*; *Abd Raman et al., 2020*) although the possibility of read depth bias was considered. Finally, it must be emphasised that these data cannot be extrapolated to other subtypes of asthma where the relationship with influenza virus, and intra-host viral variation, may be markedly different.

Nevertheless, despite these limitations these data suggest that in a mouse model of paucigranulocytic asthma there is an increased diversity of influenza viral variants some of which are associated with increased polymerase activity. Additionally, we have demonstrated that virus derived from an asthmatic host can cause more severe disease when introduced to a non-asthmatic host. Given these findings, additional studies monitoring individuals with asthma (and in particular those with non-allergic asthma) for increased influenza severity and the emergence of influenza virus variants may be of value.

## Materials and methods

### Viral strains and titrations

PVM strain J3666 stocks were prepared from mouse lung homogenate as described previously (*Garvey et al., 2005*). A/Auckland/4/2009 (Auckland/09; H1N1) stocks were prepared in embryonated chicken eggs as previously described (*Brauer and Chen, 2015*). Viral titres were determined by plaque assays on Madin–Darby canine kidney (MDCK) cells, as previously described (*Short et al., 2011*).

## Mouse strains

RAGE-deficient mice (*Ager*[-/-]) were kindly provided by Prof. Ann-Marie Schmidt, (New York University Langone Medical Centre, NY, USA). *Ager*[-/-] and C57BL/6 mice were housed in individually ventilated cages, under alternating 12 hr light/dark periods with access to clean drinking water and food *ad libitum*. All animal experiments were approved by the University of Queensland Animal Ethics Committee (permit no. 071/17).

## Establishing paucigranulocytic asthma

To model asthma development, we employed our previously described mouse model of paucigranulocytic asthma (*Arikkatt et al., 2017*). Specifically, 7-day old female and male *Ager*[-/-] mice were infected intranasally with 10 plaque forming units (PFU) of PVM in a volume of 10 μL. Thirty-five days after the primary infection, mice were reinfected intranasally with 100 PFU of PVM in a total volume of 50 μL. This model reflects the fact that asthma development is often a combination of a genetic deficiency and multiple respiratory tract infections (*Arikkatt et al., 2017*). 'Non-asthmatic' control mice were deficient in RAGE but were mock infected with 10 μL of PBS at 7 days of age.

## *In vivo* IAV infection

At 48 days of age, all *Ager*[-/-] mice were infected with 100 PFU of Auckland/09(H1N1) intranasally in 50 μL phosphate buffered saline (PBS) or mock-infected with PBS. Six-week-old female C57BL/6 mice were infected intranasally with 2000 PFU of either asthmatic host-derived IAV or non-asthmatic host-derived IAV. All infections were performed intranasally under isoflurane-induced anaesthesia using Stinger Research Aesthetic Gas Machine (Darvall, AZ, USA). Mice were monitored daily for weight loss and clinical signs of disease. At the relevant time points, mice were euthanised with intraperitoneal injection of a lethal dose of phentobarbitol (270 mg/kg) (Virbac, NSW, Australia).

## Blood oxygen saturation

Blood oxygen saturation of C57BL/6 mice was measured at 4 d.p.i. using a collar sensor and Mouseox Plus pulse oximeter (Starr, PA, USA).

## Histology and immunohistochemistry

The left lung lobe was fixed in 10% neutral buffered formalin, routine processed and embedded in paraffin, sectioned at 5 μm, and subsequently used for histology and immunohistochemistry. To identify mucus producing cells, sections were stained with Periodic Acid-Schiff (PAS) and then counterstained with Harris' hematoxylin and mounted in DEPEX. Mucus production was scored based on the percentage of mucus secreting airway epithelial cells (AECs) relative to the total number of AECs. A minimum of five airways per sample were quantified. For immunohistochemistry, sections were pre-treated with 10% normal goat serum for 30 min. Sections were then probed with anti-α-SM actin (Sigma-Aldrich, MO, USA) overnight at 4°C. Following incubation with secondary antibody, immunoreactivity was developed with Fast Red (Sigma-Aldrich) and counterstained with Mayer's haematoxylin. Airway smooth muscle (ASM) was quantified essentially as described previously (*Loh et al., 2020*). ASM was calculated as area per μm of basement membrane of small airways (defined as circumference <800 μm). Lungs were assessed for vascular changes, bronchitis, interstitial inflammation, alveolar inflammation, pneumocyte hypertrophy, and pleuritis (represented by a total score) by a veterinary pathologist who was blind to the study design. All stained slides were scanned using a digital slide scanner (Scanscope XT, Aperio Technologies, CA, USA) at 200× magnification.

## Cytokine levels

Cytokine concentrations were assessed using LegendPlex mouse anti-viral panel (Biolegend, CA, USA) according to the manufacturer's instructions.

## Quantification of influenza virus titres

Mouse lung tissues were collected in Dulbecco's minimum essential medium (DMEM; Gibco, NY, USA) and homogenised using a Qiagen Tissuelyser II (Qiagen, Hilden, Germany). The homogenate was centrifuged, and the supernatant was collected and stored at −80°C. Cell culture supernatant

was collected at indicated time points and stored at −80℃. Viral titres were measured by plaque assay on MDCK cells, as described previously (*Short et al., 2011*).

## RNA extraction, cDNA synthesis, and polymerase chain reaction (PCR)

Viral and host RNA was extracted from mouse lung samples, cDNA was synthesised, and real-time PCR was performed as described previously (*Hulme et al., 2020*). Forward and reverse primers used in the present study are supplied in *Supplementary file 1*. Gene expression was normalised relative to glyceraldehyde 3-phosphate dehydrogenase (*Gapdh*) expression and fold change was calculated using the ΔΔCt method (*Schmittgen and Livak, 2008*). Viral copy number was determined using IAV strain A/Puerto Rico/8/1934 H1N1 (PR8) virus matrix (M) gene cloned into pHW2000 plasmid and viral copy number was determined as described previously (*Short et al., 2013*).

Viral RNA for sequencing was extracted from mouse lung, and *in vitro* culture supernatant samples using the High Pure Viral RNA Kit (Roche, Basel, Switzerland) according to the manufacturer's protocol and eluted in UltraPure DNase/RNase-Free Distilled Water (ThermoFisher Scientific, MA, USA). Viral cDNA was synthesised using the Transcriptor First Strand cDNA Synthesis Kit (Roche) essentially as described previously (*Mancera Gracia et al., 2017*). Briefly, two reactions were performed per sample, using either 2.5 μM CommonUni12A primer (GCCGGAGCTCTGCAGATATCAG-CAAAAGCAGG) or 2.5 μM CommonUni12G primer (GCCGGAGCTCTGCAGATA TCAGCGAAAGCAGG). Both reactions were incubated for 10 min at 65℃ and cooled immediately on ice. RNA was reverse transcribed using Mastercycler Pro 6325 (Eppendorf, Hamburg, Germany). cDNA was pooled at a 1:1 ratio and amplified using the Phusion High-Fidelity DNA Polymerase kit (New England Biolabs, MA, USA) with 10 μM CommonUni13 primer (GCCGGAGCTCTGCAGATA TCAGTAGAAACAAGG). cDNA was amplified using Mastercycler Pro 6325 (Eppendorf). MinElute PCR Purification Kit (Qiagen) was used to purify the PCR products. To elute, unheated UltraPure DNase/RNase-Free distilled water was used instead of elution buffer. The presence of the correct size PCR products was confirmed using a 1% agarose gel containing SYBR Safe DNA Gel Stain (ThermoFisher Scientific) at 100 Volt and visualised by the Amersham Imager 600 (GE Healthcare, IL, USA).

## Next generation sequencing (NGS) for viral genes

Libraries were prepared from viral cDNA amplicons using Nextera XT DNA library prep kit (Illumina, cat no. FC-131–1024) and were deep sequenced (0.5 Gb) on an Illumina Nextseq 2000 Platform at the Australian Centre of Ecogenomics (the University of Queensland, QLD, Australia).

## NGS analysis and Shannon entropy

The haplotypes for each sample were reconstructed for each gene segment using a previously published pipeline (*Cacciabue et al., 2020*). In brief, FastQC (*Andrews, 2010*) was used for quality assurance of the NGS paired-end raw reads followed by BBtools (*Bushnell, 2014*), for removing and filtering adapters and low-quality reads. Bowtie2 (*Langmead and Salzberg, 2012*), an aligner tool to align the trimmed reads to the selected reference of the influenza strain (i.e. the inoculum), was then used. Samtools suite (*Li et al., 2009*) was used to sort, index, and generate depth and coverage statistics for read alignment files. Next, CliqueSNV (*Knyazev, 2020*) was used to infer the haplotypes and frequencies for all eight gene segments for each sample.

Shannon entropy (abundance-based diversity) was calculated using QSutils, an R package (*Guerrero-Murillo and Font, 2020*). During analysis the following assumptions were made: each paired-end read present in an alignment comes from a true viral haplotype from the original population, the occurrence of variants in a given gene segment is independent of the rest of the genes, and each haplotype in the actual population has an equal chance to be sampled.

Multiple dimension scaling (MDS) (LMDS V1.0 – R package) was utilised to visualise distance matrices to understand the dynamics of haplotype distance in a given quasi species population across donor and acceptor hosts (i.e. C57BL/6 mice infected with either 6 d.p.i. murine lung homogenate from asthmatic or non-asthmatic host) (*Jombart, 2016*). We used IRMA (iterative refinement meta-genome assembler) (*Shepard et al., 2016*) for variant calling and each of the variant in the coding sequence of PB1 gene was translated to the corresponding amino acid residue compared to the reference.

All scripts generated for this study can be found at https://github.com/akaraw/Hulme_et_al (*Karawita, 2021*) and the workflow for haplotype reconstruction can be found in *Supplementary file 3*.

Within-host alpha diversity was measured using Shannon's entropy (H):

$$H(x) = \sum_{i}^{n} P(i) \ln P(i)$$

## Identification of asthma associated mutations of IAV in clinical samples

H1N1 PB1sequences were obtained from the NIAID IRD through the website at http://www.fludb.org. The following settings were used for selecting human surveillance data following the 2009 pandemic: date range ≥2009, subtype H1N1, all countries, all sexes, and age ranges; exclude laboratory strains and duplicate sequences. Only records containing clinical metadata were returned. Using these criteria, a total of 51 human clinical samples with asthma as a medical condition were observed. Fifty-one age and sex matched human clinical samples with no medical conditions were taken for comparison. Samples were aligned using MEGA-X (version 10.1) (Pennsylvania State University, PA, USA) and the presence or absence of the mutations identified *in vivo* was determined using amino acid position.

## IAV mutagenesis

The PB1-C693Y and PB1-P651Q mutations were independently introduced into the PB1 gene segment of influenza A/Auckland/2009 (H1N1) using the Q5 Site-Directed Mutagenesis Kit (New England BioLabs) as per the manufacturer's instructions. Mutations were confirmed using the PureYield Plasmid Miniprep System Kit (Promega, WI, USA) as per the manufacturer's instructions and sequenced at the AEGRC Genotyping and Sequencing Facility at the University of Queensland. DNA was isolated using an EndoFree Plasmid Maxi Kit (Qiagen) as per the manufacturer's instructions.

## Cell culture

Human embryonic kidney 293 with large T antigen (HEK 293T) cells, MDCK cells, adenocarcinomic human alveolar basal epithelial cells (A549 cells), and Immortalized Mouse Mammary Epithelial Cells (iMMEC) were cultured in DMEM (Gibco) containing 10% foetal bovine serum (FBS; Sigma-Aldrich) and 1% penicillin streptomycin (Lonza, Basel, Switzerland) in a humidified atmosphere of 5% $CO_2$ at 37°C. All cell lines were obtained from American Type Culture Collection (ATCC; Virginia, USA) and have no mycoplasma contamination to report.

## *In vitro* viral replication kinetics

Multicycle viral replication kinetics were determined by inoculating a confluent monolayer of cells with the lung homogenate of infected mice at a multiplicity of infection (MOI) of 0.01 for 1 hr. Cells were then washed with PBS three times and fresh medium (DMEM; Gibco) was added containing TPCK-trypsin (Thermofisher Scientific). At 0, 6, 24, 48, and 72 hr post inoculation, cell culture supernatant was collected and used to determine virus titres, as described above.

## Influenza virus polymerase assay

Reporter plasmids, Fluc (0.25 µg) and pRL (0.01 µg) were co-transfected with expression plasmids encoding the AUCK/09 PA, PB1, PB2, PA, and NP (0.25 µg per plasmid) into HEK 293 T cells seeded on a 24-well plate using the calcium phosphate mediated method (*Kwon and Firestein, 2013*). Luciferase assays were performed using a dual-specific luciferase assay kit (Promega). A Renilla luciferase reporter vector was used as the internal control. Firefly luciferase activity was normalised to Renilla luciferase activity for each well. Luminescence was measured on a CLARIOstar Plus microplate reader (BMG Labtech, Ortenberg, Germany) in flat 96-white well plates (Corning, Amsterdam, the Netherlands).

## Modelling of the mutations

For K691R and C693Y the crystal structure of the PB1–PB2 subunits from IAV (A/Puerto Rico/8/1934 (H1N1)) was used as a model (PDB: 3A1G) (*Sugiyama et al., 2009*). For P651Q and S654N the structure of the influenza A bat polymerase complex (PDB: 4WSB) was used as a model (*Pflug et al.,*

*2014*). The effects of conformational stability were estimated using FoldX (version 5) (*Schymkowitz et al., 2005*) and each mutation was engineered separately. For the FoldX analysis the subroutines RepairPDB, BuildModel, and PrintNetworks were used. Images were created with PyMOL (The PyMOL Molecular Graphics System, Version 2.4.0 Schrödinger, LLC).

### Statistical analysis

Statistical analyses were performed using Graph Pad Prism software (version 9.0.1) for Windows (GraphPad Software, CA, USA). Data were tested for normality using the Shapiro–Wilk test. Where data was normally distributed, data was analysed using a two-way ANOVA (with Tukey's multiple comparison test), a one-way ANOVA (with Tukey's multiple comparison test), an unpaired Student's t-test, or a one sample t test as appropriate. Where data was not normally distributed, data was analysed using a Kruskal–Wallis test (with Dunn's multiple comparisons test), a Mann–Whitney U-test or a Wilcoxon signed-rank test as appropriate. Clinical samples were analysed using a two-sided fisher's exact test.

## Acknowledgements

SVDH was a Ph.D. fellow supported by FWO-Vlaanderen. B.V was supported by postdoctoral grant (12U7118N) of the FWO (Fonds Wetenschappelijk Onderzoek – Vlaanderen). KRU was supported by a NHMRC Career Development Fellowship (APP1130815). KRS was supported by an Australian Research Council DECRA (DE180100512). We would like to thank Professor Phillip Hugenholtz for his support in this project.

## Additional information

### Funding

| Funder | Grant reference number | Author |
| --- | --- | --- |
| Australian Research Council | DE180100512 | Kirsty Renfree Short |
| Fonds Wetenschappelijk Onderzoek | 12U7118N | Bram Vrancken |
| Fonds Wetenschappelijk Onderzoek | | Silvie Van den Hoecke |
| National Health and Medical Research Council | APP1130815 | Kyle R Upton |

The funders had no role in study design, data collection and interpretation, or the decision to submit the work for publication.

### Author contributions

Katina D Hulme, Data curation, Formal analysis, Methodology, Writing - original draft, Writing - review and editing; Anjana C Karawita, Cassandra Pegg, Helle Bielefeldt-Ohmann, Silvie Van den Hoecke, Yin Xiang Setoh, Bram Vrancken, Formal analysis; Myrna JM Bunte, Conor J Bloxham, Lauren E Steele, Nathalie AJ Verzele, Keng Yih Chew, Data curation; Monique Spronken, Resources; Kyle R Upton, Alexander A Khromykh, Supervision; Maria Sukkar, Methodology; Simon Phipps, Resources, Data curation; Kirsty R Short, Conceptualization, Resources, Data curation, Formal analysis, Supervision, Funding acquisition, Investigation, Methodology, Writing - original draft, Writing - review and editing

### Author ORCIDs

Katina D Hulme https://orcid.org/0000-0003-1322-0136
Cassandra Pegg http://orcid.org/0000-0002-6080-7047
Kirsty R Short https://orcid.org/0000-0003-4963-6184

## Ethics

Animal experimentation: All animal experiments were approved by the University of Queensland Animal Ethics Committee (permit no. 071/17).

## Decision letter and Author response

Decision letter https://doi.org/10.7554/eLife.61803.sa1
Author response https://doi.org/10.7554/eLife.61803.sa2

# Additional files

## Supplementary files

• Supplementary file 1. Forward and reverse primers used for qPCR.

• Supplementary file 2. Non-synonymous viral mutations detected in PB1 at 4 and 6 days post-influenza A virus (IAV) inoculation.

• Supplementary file 3. Workflow used to reconstruct haplotypes.

• Transparent reporting form

## Data availability

All methods and scripts used for sequencing analysis are outlined in the manuscript and supporting files. The raw sequencing data generated in this study is available on the European Nucleotide Archive with accession number PRJEB40027.

The following dataset was generated:

| Author(s) | Year | Dataset title | Dataset URL | Database and Identifier |
|---|---|---|---|---|
| Hulme KD, Karawita AC, Short KR | 2020 | A paucigranulocytic asthma host environment promotes the emergence of virulent influenza viral variants | https://www.ebi.ac.uk/ena/browser/view/PRJEB40027 | European Nucleotide Archive, PRJEB40027 |

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
