## [Decision Letter]

**Acceptance summary:**

This manuscript investigates the potential for evolution of viruses in susceptible hosts. Using a mouse model of asthma, the authors show that asthmatic animals suffer from more severe infection, but also that the virus appears to show greater evidence of evolution in asthmatic animals (based on variation in the viral sequence). These changes in the virus support more vigorous viral replication *in vitro*, suggesting an increased virulence. Crucially, infection of healthy control mice with virus derived from the asthma-prone animals also demonstrated that viral strains emerging from asthmatic animals showed increased virulence when passed back to healthy animals.

This study shows that viral evolution in compromised hosts can lead to an increase in virulence that persists when transmitted back to healthy animals. The implications of this for viral spread and evolution at a population level are important, although clearly further work is required to extend this beyond the current animal model.

**Decision letter after peer review:**

Thank you for submitting your article "A paucigranulocytic asthma host environment promotes the emergence of virulent influenza viral variants" for consideration by *eLife*. Your article has been reviewed by three peer reviewers, and the evaluation has been overseen by Miles Davenport as the Senior and Reviewing Editor. The following individuals involved in review of your submission have agreed to reveal their identity: Cody Allison (Reviewer #1); Vijaykrishna Dhanasekaran (Reviewer #2).

The reviewers have discussed the reviews with one another and the Reviewing Editor has drafted this decision to help you prepare a revised submission.

Summary:

This manuscript addresses an interesting angle of the concept that varying conditions of immunosuppression can facilitate the emergence of virulent mutant strains of Influenza. Data presented partially support this hypothesis, however there are concerns regarding the virulence of strains, deep sequence analysis, the downstream analysis of mutations, and the interpretation on influenza evolution. Therefore additional experiments are necessary to substantiate the statements in the title, Abstract and Discussion.

Essential revisions:

1) Authors state that more virulent strains emerge. While it is clear that mutational penetrance is higher in asthmatic mice, more evidence is required to establish it is more virulent.

a) weight loss and viral load are not enough to indicate virulence. This could simply be a result of the asthmatic phenotype, being unable to control virus. The increased replication *in vitro* is insufficient to ascribe to a mutant with higher replicative capacity.

b) what are the predominant mutations growing out of the cell lines following incubation with lung homogenates? Does it vary between the mice? If they're replicating faster they should be the predominant clones. Were the 3 asthmatic mice used for this experiment selected based upon the frequency of mutations identified in them?

c) should have histology and lung cytokines from asthmatic and non-asthmatic mice following challenge to corroborate increased virulence.

d) what happens to non-asthmatic mice when you challenge them with virus isolated from homogenised lung from the asthmatic group? Non-asthmatic animals should develop symptoms (weight loss, cytokines, oedema, infiltrates) if indeed you are generating strains of greater virulence.

2) Shannon's entropy scores in Figure 3 are non-parametric, with many asthmatic animals giving a score of zero. For PB1 (or any of the segments), can entropy scores be correlated with individual mice across the ~2-log spread of viraemia in this group (in Figure 2B) (i.e. does increased viral diversity correlate with high or low viraemia in those animals)?

3) There is concern with the use of a single measure of genetic diversity (Shannon entropy), which has been shown repeatedly to produce biased estimates (Please see. Zhao and Illingworth, Virus Evolution 5:vey041 https://doi.org/10.1093/ve/vey041 ). It is also not clear what the effect of the initial template load on measures of genetic diversity at each site/gene. Did the authors check if there was sufficient template diversity as a low diversity can bias all estimates? Additionally, it would be important to show any variation between samples / days as supporting information.

4) As before, since the rest of the paper relies heavily on single mutations (e.g. PB1 693), it would be important to use robust methods, or multiple methods to check whether they are consistent.

5) A major goal of this study was to address the importance of co-morbidities is on influenza evolution, but this hypothesis is not directly tested in this paper. Using C693Y as example, it is not clear what their prevalence is among naturally circulation H1N1pdm09 viruses. By filtering publicly available sequences, the authors have identified sequences from 51 asthma cases and compared these to 51 cases matching age and sex with no co-morbities (randomly I assume), but I feel these number are quite low to derive meaningful comparisons. To make inferences about the effect on influenza evolution, I suggest to map important mutations onto gene phylogenies to identify the dominance of mutations in naturally circulating populations. e.g. has Y693 or other PB1 mutations fixed in any of the circulation sub-lineages of H1N1.

It is also interesting this mutation is predicted to increase stability, and that they potentially show increased polymerase activity. If correct, shouldn't this mutation be dominant among naturally circulating strains?

6) As a part of the RdRp, mutations within PB1 gene by itself can increase mutational frequency, but this is not explored/discussed in light of results presented.

---

## [Author Response]

Essential revisions:1) Authors state that more virulent strains emerge. While it is clear that mutational penetrance is higher in asthmatic mice, more evidence is required to establish it is more virulent.a) weight loss and viral load are not enough to indicate virulence. This could simply be a result of the asthmatic phenotype, being unable to control virus. The increased replication *in vitro* is insufficient to ascribe to a mutant with higher replicative capacity.

We acknowledge that weight loss and viral load alone are not enough to indicate virulence. We have now updated the figure legend of Figure 2 to reflect a move away from virulence and instead focus on disease severity “RAGE-deficient asthmatic mice experience more severe influenza than non-asthmatic mice”. Furthermore, we have included day 4 viral titre data, histology scores of all time points, and cytokine and inflammatory marker data in the updated Figure 2. We have subsequently updated the following sections of the manuscript:

Materials and methods:

“Viral and host RNA was extracted from mouse lung samples, cDNA was synthesised, and real time PCR was performed as described previously (Hulme et al., 2020). […] Viral copy number was determined using IAV strain A/Puerto Rico/8/1934 H1N1 (PR8) virus matrix (M) gene cloned into pHW2000 plasmid and viral copy number was determined as described previously (Mancera Gracia et al., 2017).”

Materials and methods:

“Lungs were assessed for vascular changes, bronchitis, interstitial inflammation, alveolar inflammation, pneumocyte hypertrophy, and pleuritis (represented by a total score) by a veterinary pathologist who was blind to the study design.”

Results:

“While there was an increase in the histopathology score of mock infected asthmatic mice compared to non-asthmatic, there was no observable difference following IAV infection (Figure 2C). Furthermore, asthmatic mice showed a significantly impaired type I interferon response with a down regulation of both IFN-α and interferon inducible MX1 following IAV infection (Figure 2D).”

b) what are the predominant mutations growing out of the cell lines following incubation with lung homogenates? Does it vary between the mice? If they're replicating faster they should be the predominant clones. Were the 3 asthmatic mice used for this experiment selected based upon the frequency of mutations identified in them?

The question of mutations arising *in vitro* during the replication kinetics experiment is an interesting one. It is important to note that the asthmatic and non-asthmatic mice used for the replication kinetics were chosen at random to avoid any bias. In response to the reviewer’s suggestion, we have since deep sequenced the PB1 genome segment produced from the *in vitro* MDCK experiments and compared them back to their inoculum parent sequence to determine the predominant mutations above a 5% threshold.

One of the parent isolates was missing sufficient reads in regions of interest and was excluded from the analysis. Of the two remaining isolates, there was no change in the consensus sequence of the daughter population after *in vitro* replication compared the consensus sequence of the parent. In terms of minority viral populations, in one daughter population no viral variants (with non-synonymous mutations) were detected that constituted between 5-49% of the viral population. In the second daughter population, one non-synonymous mutation (PB1 I205M) was detected that constituted 28% of the viral population. However, this mutation was present in consensus sequence of the parent isolate, albeit at a higher percentage (95%). Together, these data suggest that the increased replication of viral isolates from asthmatic mice was not associated with a large number of additional *in vitro* adaptations. We have subsequently updated the manuscript to include these results the following section:

Results:

“Interestingly, comparing two of the viral samples pre and post *in vitro* replication in MDCKs we did not detect a change in the consensus sequence of the daughter population after *in vitro* replication compared the consensus sequence of the parent. In terms of minority viral populations, in one daughter population no viral variants (with non-synonymous mutations) were detected that constituted between 5-49% of the viral population. […] Together, these data suggest that the increased replication of viral isolates from asthmatic mice was not associated with a large number of additional *in vitro* adaptations”.

c) should have histology and lung cytokines from asthmatic and non-asthmatic mice following challenge to corroborate increased virulence.

Please see our response to reviewer query 1a above, we have now included histology and lung cytokine data.

d) what happens to non-asthmatic mice when you challenge them with virus isolated from homogenised lung from the asthmatic group? Non-asthmatic animals should develop symptoms (weight loss, cytokines, oedema, infiltrates) if indeed you are generating strains of greater virulence.

In response to the reviewer’s query, we have now performed experiments whereby we inoculated non-asthmatic C57BL/6 mice with lung homogenate from either asthmatic or non-asthmatic mice at a PFU of 2000 PFU per mouse (**Figure 8**). Interestingly but unsurprisingly, we saw that asthmatic host-derived IAV was associated with significantly more weight loss (Figure 8A), lower blood oxygen saturation (Figure 8B), and higher viral load (Figure 8C) than non-asthmatic host-derived IAV. Furthermore, mice infected with asthmatic host derived IAV had a significant upregulation in various anti-viral and pro-inflammatory cytokines (Figure 8E). Furthermore, upon infection of a non-asthmatic host with non-asthmatic derived IAV, there is an observable divergence in the PB1 haplotype differences as the virus continues to adapt to its host (Figure 9A). By contrast, upon infection of a non-asthmatic host with asthmatic derived IAV, there is a consolidation of the PB1 mutations that arose during the initial infection (Figure 9B). Together this data suggests that asthmatic hosts could facilitate the emergence of new strains that have increased pathogenicity.

We have subsequently updated the Materials and methods, Results and Discussion:

Materials and methods:

“Mouse strains

RAGE-deficient mice (*Ager*^-/-^) were kindly provided by Prof. Ann-Marie Schmidt, (New York University Langone Medical Centre, USA). […] Blood oxygen saturation of C57BL/6 was measured at 4 days post-infection using a collar sensor and Mouseox Plus pulse oximeter (Starr, Oakmont, PA, USA)”.

Materials and methods:

“Multiple dimension scaling (MDS) (LMDS V1.0 – R package) was utilised to visualise distance matrices to understand the dynamics of haplotype distance in a given quasi species population across donor and acceptor hosts (i.e. C57BL/6 mice infected with either 6 d.p.i murine lung homogenate from asthmatic or non-asthmatic host) [Shepard et al., 2016]. […] All scripts used for this study can be found at https://github.com/akaraw/Hulme_et_al and the workflow for haplotype reconstruction can be found in Supplementary file 3.”

Results:

“Asthmatic host-derived IAV exhibits greater disease severity and viral diversity upon infection of a wild-type host

Vulnerable patient groups may be a source for the emergence of new viral strains [Honce et al., 2020]. […] Together this data suggests that asthmatic hosts could facilitate the emergence of new strains that have increased pathogenicity”.

Discussion:

“Further to this, non-asthmatic host-derived viruses showed host adaption and little sign of morbidity in wild-type mice, while asthmatic-host derived virus resulted in increased morbidity. This suggests that viral variants born from an asthmatic host could (transmission permitting) lead to the emergence of more pathogenic viral strains”.

2) Shannon's entropy scores in Figure 3 are non-parametric, with many asthmatic animals giving a score of zero. For PB1 (or any of the segments), can entropy scores be correlated with individual mice across the ~2-log spread of viraemia in this group (in Figure 2B) (i.e. does increased viral diversity correlate with high or low viraemia in those animals)?

This is an interesting query and we thank the reviewer for suggesting it. To answer this, we assume the reviewer is referring to the viral load in the lung (PFU) when they are speaking of “viraemia” and we will answer accordingly. For the correlation analysis, we focused on the PB1 genome segment as this is the only gene that showed a difference between asthmatic and non-asthmatic in our results (Figure 3). On an individual animal level, we can see a positive relationship within the D6 asthmatic group, between the weight-loss (as a measure of disease severity) and both the Shannon Entropy (Author response image 1), and the total number of mutations in the PB1 genome segment (Author response image 1). However, no correlation between the viral load and Shannon Entropy (Author response image 1), or the total number of mutations in the PB1 genome segment (Author response image 1) was detected. Together this suggests that at day 6, asthmatic mice that experienced more severe influenza (as determined by weight-loss) produced more viral variants, but that this did not correlate with an increased viral load.

We are happy to include this data as supplementary material if the reviewers believe it adds value to the paper.

**Author response image 1. respfig1:** Asthmatic day 6 samples show a positive relationship between weight loss and measures of viral diversity in the PB1. Influenza viral RNA isolated from the lung tissue of asthmatic (n=27) and non-asthmatic (n=25) mice was analysed for viral variants. Weight-loss as a function of (A) Shannon entropy of PB1 genome segment and of (B) SNVs detected in the PB genome segment. Viral titre (PFU/mL) as a function of (C) Shannon entropy of PB1 genome segment and of (D) SNVs detected in the PB genome segment. Samples with a Shannon Entropy value of 0 were excluded. A simple linear regression was fit, and statistical significance was determined using Prism. * p ≤ 0.05.

3) There is concern with the use of a single measure of genetic diversity (Shannon entropy), which has been shown repeatedly to produce biased estimates (Please see. Zhao and Illingworth, Virus Evolution 5:vey041 https://doi.org/10.1093/ve/vey041 ). It is also not clear what the effect of the initial template load on measures of genetic diversity at each site/gene. Did the authors check if there was sufficient template diversity as a low diversity can bias all estimates? Additional it would be important to show any variation between samples / days as supporting information.

We agree with the reviewers that one measure of genetic diversity is limiting. However, we selected this approach to be consistent with recently published works in this field, which have relied heavily on Shannon entropy without any other measures of diversity (please see Honce et al., 2020). In light of the reviewer’s concerns, we have now compared the read depth between samples and can conclude that no bias based on difference in read depth are present. We have now included the read depth results as Figure 3—figure supplement 1, and have included the use of a single measure of genetic diversity as a limitations of the study. These additions correspond to changes in the following sections:

Results: “To account for any bias from difference in sequencing coverage, the read count for each segment was analysed and there were no significant differences between groups (Figure 3—figure supplement 1)”.

Discussion:

“Additionally, only a single measure of genetic diversity was used (Shannon entropy), however we took into account any read depth bias, and within the field of influenza diversity this method alone has been routinely used [Honce et al., 2020; Abd Raman et al., 2020].”

We would also like to remind the reviewers that all the viral sequencing data generated in this study is available on the European Nucleotide Archive with accession number PRJEB40027 for anyone who wishes to look at our data further.

4) As before, since the rest of the paper relies heavily on single mutations (e.g. PB1 693), it would be important to use robust methods, or multiple methods to check whether they are consistent.

We would like to stress to the reviewers that our paper does not rely on any one mutation for our conclusion. Rather, we are suggesting that there is an observable increase in diversity of virus isolated from an asthmatic host, compared to a non-asthmatic host. Additionally, we have now demonstrated that virus derived from an asthmatic host can cause more severe disease when introduced to a non-asthmatic host (see response to reviewer query 1d above). Given these findings, additional studies monitoring individuals with asthma (and in particular those with non-allergic asthma) for increased influenza severity and the emergence of influenza virus variants may be of value.

The purpose of creating the C693Y mutation was to demonstrate that mutations found in key areas of the PB1 protein structure, exclusive to the asthmatic group, could affect (specifically, increase) the replication capability of the virus. However, to alleviate concerns that the paper relies heavily on a single mutation (C693Y), we have now performed mutagenesis and introduced the P651Q mutation into the PB1 plasmid and observed polymerase activity. These results confirm the crystal structure findings and indicate that the P651Q mutation increases polymerase activity compared to the wild type.

We have included the data in Figure 7 and a discussion of these points in the following sections:

Abstract:

“Specifically, C693Y and P651Q mutations in the PB1 of influenza virus was exclusively detected in mice with the cardinal features of paucigranulocytic asthma and these were associated with increased viral replication and polymerase activity”.

Results:

“PB1 C693Y and P651Q increases viral polymerase activity

We next sought to investigate the individual effect of the PB1 C693Y and P651Q mutations on viral polymerase activity using a minigenome assay, given its their predicted increase in stability. […] Together these data suggest that the C693Y and P651Q mutations identified exclusively in asthmatic hosts, is altering the interaction between PB1 and PB2 in a stabilising manner, resulting in increased polymerase activity”.

Discussion:

“Our structural analysis of PB1 P651Q predicted a stabilising effect, which was consistent with an increase in polymerase activity relative to the wild type”.

Materials and methods:

“The PB1-C693Y and PB1-P651Q mutations were independently as introduced into the PB1 gene segments of influenza A/Auckland/2009 (H1N1) using the Q5 Site-Directed Mutagenesis Kit from New England BioLabs as per the manufacturer’s instructions”.

5) A major goal of this study was to address the importance of co-morbidities is on influenza evolution, but this hypothesis is not directly tested in this paper. Using C693Y as example, it is not clear what their prevalence is among naturally circulation H1N1pdm09 viruses. By filtering publicly available sequences, the authors have identified sequences from 51 asthma cases and compared these to 51 cases matching age and sex with no co-morbities (randomly I assume), but I feel these number are quite low to derive meaningful comparisons. To make inferences about the effect on influenza evolution, I suggest to map important mutations onto gene phylogenies to identify the dominance of mutations in naturally circulating populations. e.g. has Y693 or other PB1 mutations fixed in any of the circulation sub-lineages of H1N1.It is also interesting this mutation is predicted to increase stability, and that they potentially show increased polymerase activity. If correct, shouldn't this mutation be dominant among naturally circulating strains?

The reviewer is correct in the assumption that the 51 cases were randomly selected. We also agree with the reviewers that the sample size of 51 clinical cases is quite small. Unfortunately, at the time of manuscript preparation, there were only 51 clinical samples that listed asthma as a co-morbidity, thus limiting our analysis.

We agree with the reviewers that in order to make inferences about the effect on influenza evolution we would need to map the mutations onto phylogenies. However, as we used a mouse model, we do not expect to see these exact asthmatic mouse-specific mutations in the human population. Rather than suggesting that the mutations found within this study could arise in naturally circulating population, we are suggesting that asthmatic hosts give rise to more pathogenic mutations and using the clinical samples to highlight a similar trend.

6) As a part of the RdRp, mutations within PB1 gene by itself can increase mutational frequency, but this is not explored/discussed in light of results presented.

We thank the reviewer for this comment. Due to biosafety limitations, we cannot introduce the mutations that were identified into an infectious virus as this would be classified by our institution as gain-of-function work. As a result, we cannot experimentally investigate if the C693Y or P651Q mutations lead to increased mutation rates. However, previous studies have indeed shown that specific mutations in the PB1 gene segment can contribute to altering the fidelity of the RdRp (Lin, et al., 2019) and as such the emergence of mutations.

We have now included a discussion of this possibility in the Discussion:

“It is well established that SNVs can drastically change the replication fidelity and pathogenicity of a virus [Lin et al., 2019; Pauly et al., 2017]. […] Nevertheless, our results should be taken as proof of principal that an asthmatic host is more likely to produce more virulent viral variants – a finding that holds wide consequences for the emergence of new viral strains”.